# Magnetic and electric field accelerate Phytoextraction of copper *Lemna minor* duckweed

Natalia Politaeva<sup>☉</sup>, Vladimir Badenko[ID]*<sup>☉</sup>

Civil Engineering Institute, Peter the Great Saint Petersburg Polytechnic University, Saint Petersburg, Russian Federation

☉ These authors contributed equally to this work.

* badenko_vl@spbstu.ru

**Data Availability Statement:** All relevant data are within the manuscript.

**Funding:** The research is partially funded by the Ministry of Science and Higher Education of the Russian Federation as part of World-class

## Abstract

In accordance with the opinion of the World Health Organization and the World Water Council the development of effective technologies for the treatment of wastewater from heavy metals for their discharge into water bodies or reuse is an urgent task nowadays. Phytoremediation biotechnologies is the most environmentally friendly and cheapest way of the treatment of wastewater, suitable for sustainable development principals. The main disadvantage of the phytoremediation is the slow speed of the process. A method for accelerating the process of phytoremediation by the combined effect of magnetic and weak electric fields is proposed. The purpose of this study is to determine the values of the parameters of the magnetic and weak electric fields that are most suitable for extracting cuprum ions from wastewater using the higher aqua plants (*Lemna minor*). A corresponding technological process based on the results of the study is proposed. The results have shown that the removal of copper cations from sulfate solutions effectively occurs in the initial period of time (1–5 hours) under the influence of a magnetic field with an intensity of $H = 2$ kA/m. Under the combined influence of an electrical current with density $j = 240$ µA/cm$^2$ and a magnetic field ($H = 2$ kA/m) the highest rate of copper extraction by duckweed leaves is achieved. Under these conditions, the greatest growth and development of plant leaves occurs. The paper presents the results of determining of the parameters of the electrochemical release from the eluate of the spent phytomass of duckweed. It has been determined that the release of metal occurs at $E = 0.32$ V. An original scheme for wastewater treatment from copper with subsequent separation of copper from the spent phytomass of duckweed is proposed. In general, the presented results are a scientific justification of wastewater treatment technologies and a contribution to resolving the crisis in the field of fresh water supply. An important contribution in the circular economy is a technology recommendation proposed for recovering copper from duckweed after wastewater treatment.

Research Center program: Advanced Digital Technologies (contract No. 075-15-2020-934 dated 17.11.2020). The funders had no role in study design, data collection and analysis, decision to publish, or preparation of the manuscript.

**Competing interests:** The authors have declared that no competing interests exist.

## 1. Introduction

The growth of the world's population, industrialization, urbanization and the intensification of agricultural production have led to a crisis in water supply and an escalation of problems associated with fresh water resources and their pollution, especially in developing countries [1,2]. Clean water demand and the number of people experiencing a scarcity of water resources are constantly growing. This problem is pointed out by international agencies such as the World Health Organization (WHO) and the World Water Council [3,4]. In this connection, the urgent task is to develop effective technologies for wastewater (WW) treatment, for their subsequent discharge into water bodies or reuse according to the principals of the Cleaner Production Programme [2,5,6].

Heavy metal ions, which are usually present in industrial and agricultural WW, are especially toxic [7]. Heavy metals (such as *Pb*, *Zn*, *Hg*, *Cd*, *Cr*, *As*, *Cu*, *Mn*, *Fe*, *Ni*, *Co*) entering a human body can cause serious diseases and have the property of accumulating in the food chain, they are resistant to biodegradation and they are usually stable in the environment [8]. However, on the other hand, heavy metals such as copper, zinc, boron, molybdenum are necessary for the growth and development of animals and plants, but they are harmful when their concentrations exceed permissible limits [9]. Therefore, various regulatory agencies, both global–WHO and national (US EPA—United States Environmental Protection Agency) level install regulation for the acceptable limit of heavy metals in drinking water to rule out possible health problems [10–12].

Urban, agrarian and industrial WW treatment technologies can be classified into physical, chemical and biological methods, each of which have its own advantages and disadvantages [13]. Heavy metals are removed from WW by methods such as membrane filtration, chemical precipitation, adsorption, chelation, and ion exchange [14,15]. It should also be noted that heavy metals are widely used in modern technologies and the demand for heavy metals is constantly growing. Therefore, their recovery from WW for reuse can bring not only environmental, but also economic benefits, which is the way to a circular economy and sustainable development [16,17]. The analysis shows that despite recent advances in water purification (ion exchange, photocatalysis and adsorption processes) most existing methods for water treatment have high operating costs and limitations in terms of the efficiency of removing contaminants [18]. For example, the adsorbents used in these technologies are relatively expensive and therefore not always available in developing countries, where the problem of clean drinking water is especially acute [1,19,20].

Compared to physical and chemical technologies, biotechnology costs are lower for WW treatment in real situation and more suitable for ongoing environmental restriction and sustainable developments [21]. Recently, especially in warm countries, the technology of purification of WW, well known as phytoremediation, has been intensively introduced with the help of higher aquatic plants [22,23]. The cells of higher aquatic plants are able to efficiently extract and utilize pollutants. Phytoremediation is the most environmentally friendly and cheapest way to clean water [24–26]. The dead and depleted plants (phytosorbents) are non-toxic and can be used as additives to animal and bird feeds to produce compost, vermicompost and biodiesel [27,28].

The term *phytoremediation* (*phyto*—plant and *remediation*—redress) has not a very long history. It was first used by the American scientist I. Raskin in 1994 [29]. Higher aquatic plants in reservoirs perform the following main functions, which are of the greatest importance for environmental sustainability: *filtration* (contribute to sedimentation of suspended solids); *absorbing* (absorption of nutrients and some organic substances); *cumulative* (the ability to accumulate some metals and organic substances that are difficult to decompose); *oxidative*

(water is enriched with oxygen in the process of photosynthesis); *detoxification* (plants are able to accumulate toxic substances and convert them to non-toxic) [25].

The expansion of the use of phytoremediation technologies is observed in all countries of the world [30]. For example, structures with wetland vegetation for treatment of domestic WW in Ireland [31], the Netherlands [32], China [33] are described. In China, phytoremediation technologies are used for both soil and polluted water treatment [22]. Such technologies are also widely used in India [34]. For example, the bioaccumulation rate in the roots and shoots of natural vegetation *Typha angustifolia* and *Echhornia crassipus* in the urban lake Lakshmi Taal, WW from the city of Jhansi in Central India was studied [35]. The possibility of using microalgae for the treatment of WW and natural waters from a wide range of pollutants, both organic (nitrates, sulfates) and inorganic (heavy metals), has been studied by many scientists because of their circular bio economy perspective [36]. *Chlorella sorokiniana* microalgae are widely used in the phytoremediation technologies [37]. In [38], a comprehensive work was carried out to study the treatment of urban WW using green algae–*Chlorella. sorokiniana* and *Scenedesmus obliquus*. In [39] authors investigated the extraction of heavy metal ions such as copper, nickel, cadmium, from drinking water by *Chlorella sorokiniana*. The processes of sorption of *Pb*, *Cu*, and *Cd* using green algae were studied in [40]. The microalgae *Chlorella sorokiniana*, *Chlorella vulgaris*, and *S. obliquus* are used as a sorbent for extracting drugs from pharmaceutical WW. For example, paracetamol is most effectively extracted by *Chlorella sorokiniana* strain [41].

Aquatic plants have an extensive root system that helps them accumulate pollutants in their roots and shoots [42]. The cultivation of aquatic plants is time-consuming, that may limit the growing need for phytoremediation [43]. Nevertheless, this disadvantage is replaced by a number of advantages that this WW treatment technology possesses [44,45]. Various higher aquatic plants are used as biosorbents [46]. Such plants must satisfy certain requirements: tolerance to toxicants and possible temperature changes, insect action, halophilicity, i.e. tolerance to increased salt concentrations, drought, possible stress factors, the presence of enzymes that ensure degradation toxicants, the ability to accumulate large quantities of inorganic toxicants in the intracellular space [47].

The most common plant-phytoremedants are eichhornia [48] and duckweed [49]. *Eichornia crassipes* (Water hyacinth) is a representative of higher aquatic vegetation. It is known that *Eichornia* is able to transform complex high-molecular and low-molecular compounds into simpler ones. *Eichornia* reaches the degree of water purification from its compounds: nitrogen (up to 70%), phosphorus (up to 60%) and total organic carbon (up to 60%) [50]. Duckweed is a free-floating aquatic plant that grows on the surface of the water. Duckweed belongs to the *Araceae* family (the *Lemnoideae* subfamily) and consists of five genera, in which at least 40 species have been identified [51]. Duckweed can survive at different *pH* (3.5 to 10.5) and temperatures from 7 to 35˚C), so it is widely used in phytoremediation [52]. Duckweed can eliminate a huge amount of various heavy metals, inorganic and organic pollutants, pesticides, nutrients which come from agricultural WW, industrial and domestic WW, also it absorbs ammonia [43]. The nitrogen content in the cells of duckweed (trifoliate duckweed, *Lemna trisucula*, *Lemna minor*, etc.) may exceed its concentration in water by 2000 times, phosphorus—7000, potassium—5000 times [53]. These properties of duckweed made it possible to use them for the purification of industrial and household water all over the world [54].

The main disadvantage of phytoremediation processes is the low speed of purification [7,55]. To accelerate the processes of phytoremediation, various physical factors, the influence of which depends on wavelength, frequency of oscillations of electromagnetic waves, power and time of exposure [49,50,56], are used. For example, it is known that the effect of a magnetic field can manifest itself either as a stimulant or as a retardant for the development of cells and

plant root systems [57] and, as a consequence, affect the phytosorption of heavy metals by plants during phytoremediation of polluted water bodies. t should be noted that a constant magnetic field (MF) has different effects on the growth, development and evolution of plants [58]. This influence was first investigated by the Russian scientist P.V. Savostin in 1930. In addition to magnetic fields [59], other influences were used to speed up the phytoremediation process, including: laser impact, ultraviolet and infrared radiation [49,50]. It was shown that their effect on plants is associated with a change in the potential of cell membranes, an increase in their permeability and, as a consequence, an acceleration and completeness of cation absorption. Cell membranes, as a natural barrier, are the first to be exposed to physical factors with ability to quickly respond to deviations in the environmental conditions. However, the changes that occur in the membranes entail a cascade of shifts in the metabolism of the entire cell. The permeability of membranes increases, depolarization of the membrane potential of the plasma membrane occurs, the *pH* (acidity) of the cytoplasm shifts to the acidic side, the activity of hydrogen ions $H^+$ increases [60].

Copper (*Cu*) is one of the toxic heavy metals which are widely used in modern industry, for example, in electronic chips, batteries, cell phones, etc. On the other hand, *Cu* is an essential element for human beings for healthy development, its quality is essential in the creation of hemoglobin in red platelets [7]. But high dosages of copper can be to a great degree dangerous when it enters the human body through food, residue and water, and the usage of copper-polluted water or food can cause different dangerous diseases [61]. For this reason, the results of the research on methods of accelerating phytoremediation for purifying WW from copper ions presented in this paper are relevant in the context of a sustainable development and cleaner production. A lot of works are devoted to the influence of one physical factor, especially, a magnetic field, on phytoremediation processes [2,9,16,24,42,57–59]. The influence of the combined action of physical factors on the rate of extraction of pollutants is poorly studied. The objective of this research is to study the influence of the magnetic field and the combined effects of magnetic and weak electric fields on the process of copper extraction using the higher aquatic plant *Lemna minor*. On this base a specific technology has also been presented.

## 2. Materials and methods

The higher aquatic plant of the duckweed *Lemna minor* was used as an object of phytoremediation research. At first, the influence of a magnetic field (MF) of various strengths (*H* = 0.0; 0.5; 1.0; 2.0; 4.0 kA/m) on copper extraction by duckweed *Lemna minor* was analyzed. Plants of the same maturity and weight, 20 g/l, were planted in model solutions with a $Cu^{2+}$ cation concentration of 1 mg/L and exposed to a constant MF in a specially designed Magnit-1 device (manufactured in Russia) (Fig 1). $CuSO_4$ was used to prepare a model solution. After 1, 2, 3, 4, 5, 24, 48, 120, 144, 168, 240 hours a residual concentration of copper cations was measured. On the next stage, to study the combined action of two factors, a magnetic field with an intensity *H* = 2 kA/m and a weak electric field with a current density *j* = 80, 240, 480 μA/cm$^2$ were used. To create weak electric fields, an electrochemical system with a volume of 1 liter, in which an aluminum foil was applied as a cathode and a graphite rod was applied as an anode, was used. A weak electric field on the electrodes and the potential on the electrodes were maintained by using an IPC Compact potentiostat (manufactured in Russia).

An electrochemical cell with a duckweed in a CuSO4 solution (initial concentration of $Cu^{2+}$ *Ci* = 1 mg/L) was placed in a magnetic field and held for different times. After 1, 2, 3, 4, 5, 24, 48, 120, 144, 168, 240 hours the residual concentration of copper cations was measured.

The final concentration of copper ions was determined by the voltammetric method on a robotic complex "Expertise VA-2D" with an electrode "3 in 1" (manufactured in Russia).

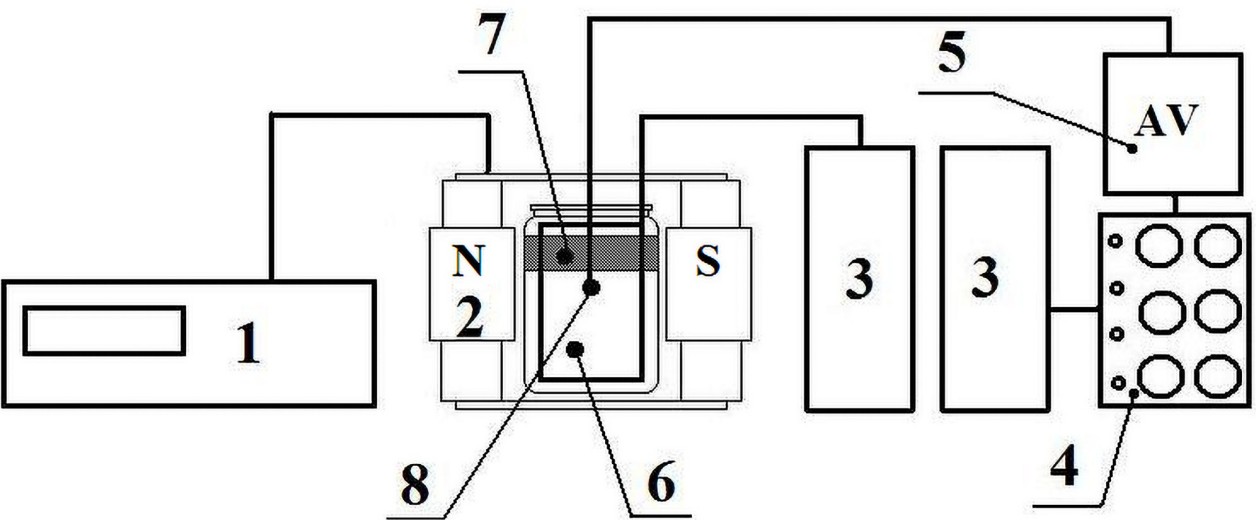

**Fig 1. Scheme of the environment for experiments.** 1 –a constant current source; 2 –magnetic coil; 3 –nickel-cadmium batteries; 4 –a resistance store; 5 –a voltammeter; 6 –an aluminum electrode; 7 –duckweed; 8 –a graphite electrode.

Natural resources are used to obtain pure metals and their extraction is harmful to the environment. The use of metals from the recycled resources is economically beneficial and environmentally sound. Therefore, we studied the possibility of extracting copper from the spent phytomass of duckweed by the electrochemical (potentiostatic) method. This corresponds to the rules of the circular economy, where waste material (duckweed phytomass) becomes a raw material for a new product (pure copper).

To recover copper from the spent biosorbent, an eluate was prepared using concentrated sulfuric acid. Then the potentiostatic method was used to determine the copper content in the eluate. The potentiostatic method was used to obtain a series of potentiostatic curves in the extraction of copper from the eluate at a constant potential. Copper was precipitated at the cathode at potentials ($E = 0.30; 0.32; 0.34$ V) close to the potential for copper precipitation. According to the laws of electrolysis by M. Faraday, we can calculate the mass of the substance released on the electrode $m$, in grams:

$$m = Q M / (F z), \tag{1}$$

where: $Q$ is the total electric charge passed through the substance in pendants, $F = 96.485$ C/mol is the Faraday constant, $M$ is the molar mass of the substance in g/mol, $z$ is the valence number of ions in the substance (electrons are transferred to the ion). In a simpler applicable form for the current case:

$$m = q I t, \tag{2}$$

where $q = 1.185$ g/(Ah) is the electrochemical equivalent of $Cu^{2+}$, in the simple case of direct current electrolysis $I$ (Ah).

Potentiostatic measurements were carried out at room temperature on an IPC Compact potentiostat (manufactured in Russia) with the recording of the experimental results on a self-recording potentiometer.

Graphite rods served as working and auxiliary electrodes (the auxiliary electrode area was ~ 40 times larger than the working area). A normal silver chloride electrode in *KCl* solution was used as a reference electrode. During the electrochemical measurements a sealed three-

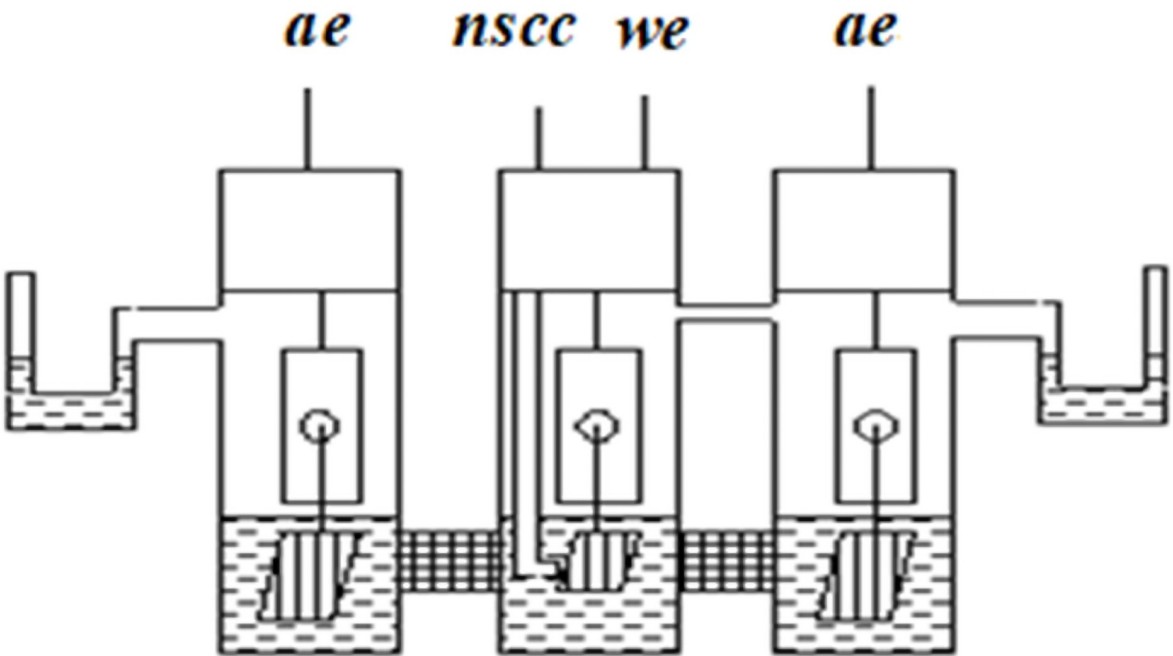

**Fig 2. Three-electrode cell with separated Schott filters with cathode and anode spaces.** ae–auxiliary, we–working, nscc–normal reference electrodes with silver chloride.

electrode cell with an electrolytic gate and separated cathode and anode spaces was used by applying Schott filters (Fig 2), which prevented mixing of the reaction products formed in the near-electrode layer.

Microstructural studies were performed on a SteREO Lumar fluorescent stereomicroscope. V12 of the German company Karl Zeiss with a smooth change in magnification ratio of 1:12. and on a scanning electron microscope brand JSM-7001F, Jeol. For all 3 parallel experiments, the arithmetic mean and deviation were determined. It was found that critical value of Student's criterion = 95%.

## 3. Results and discussion

At the first stage the influence of a constant magnetic field on phytoremediation processes was studied. When studying the effect of a magnetic field of different strengths (Fig 3) it was found that the removal of copper cations from sulfate solutions proceeds most efficiently in the initial period of 1–5 hours.

High metal removal rates are achieved up to ~ 5 days. The mechanism of a bio-sorption by a plant of toxicants is carried out by the movement of dissolved substances through the cell membrane of plants. It can be seen in Fig 3 that under the influence of a magnetic field with a strength of 2 kA/m *Lemna minor* duckweed absorbs copper ions with greater speed and intensity than without a magnetic field and with an intensity $H = 0.5; 1.0; 4.0$ kA/m. A 2 kA/m magnetic field exerts its effect as a stimulating factor. This fact is explained by the increase in the permeability of the plant cell membrane and the stimulation of a plant growth.

Therefore, at the next stage we studied the effect of MF on the growth and reproduction of *Lemna minor* duckweed. It was found that the greatest growth and development of *Lemna minor* duckweed occurred without MF and when influenced by MF $H = 2.0$ kA/m (Table 1). MFs with a strength of $H = 0.5; 1,0, 4$ kA/m influenced on the duckweed depressingly.

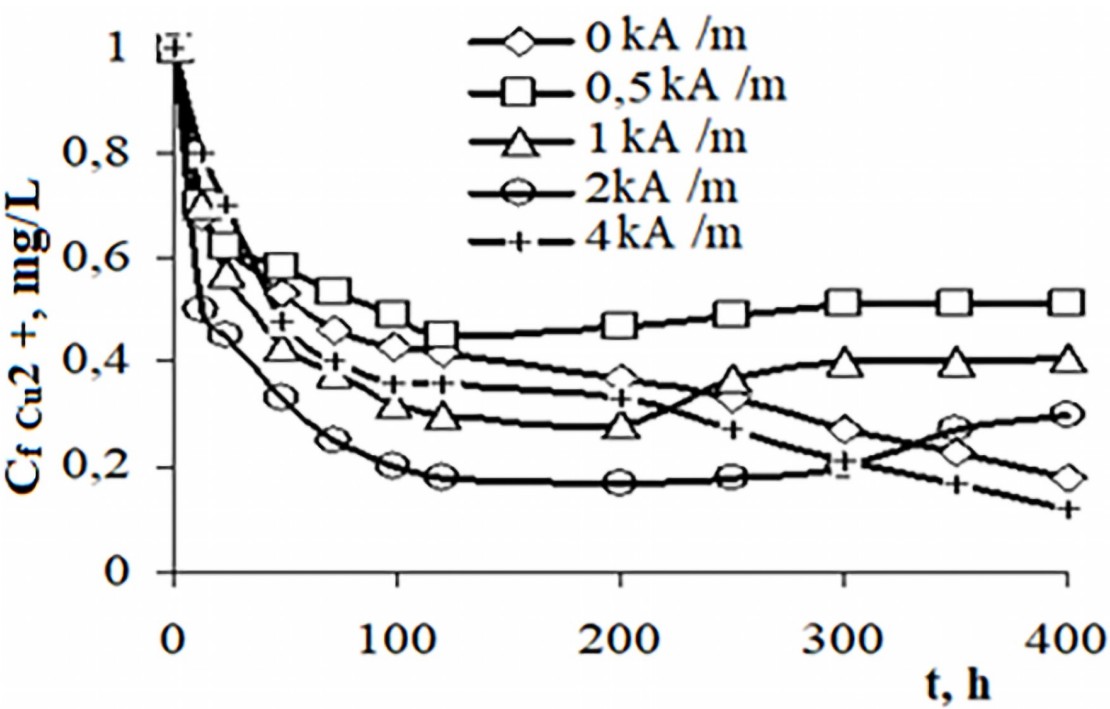

**Fig 3. Comparative results on the effect of MF of different strengths on the process of the copper extraction by *Lemna minor* duckweed.** $t$–time (h), $Ci$ is initial concentration $Cu^{2+}$ mg/L = 1 mg/L, $C_f$ Cu2 + is concentration of $Cu^{2+}$, mg/L.

Upon desorption by the plants of the accumulated excess amount of copper, a change in their appearance was observed. The initial (natural) state of *Lemna minor* is shown in Fig 4. On the second day under the influence of a magnetic field with a strength of $H = 0.5$ kA/m the duckweed changed its color from green to light green, it turned yellow on the 4th-5th day, necrosis was observed on the 10th day and then *Lemna minor* died. Under the influence of a magnetic field with a strength of $H = 1.0$ kA/m a change in the color of the plant from green to light green with a yellowish tint was also observed on the 2nd day, on the 4th day the duckweed completely turned yellow and on the 10th day the plant died. A somewhat better result was observed under the influence of a magnetic field of $H = 2.0$ kA/m. On the second day a discoloration of the leaves from green to light green with a yellow color was observed, on about the

**Table 1. Dynamics of *Lemna minor* duckweed breeding with and without MF.**

| Time, hours | The number of duckweed leaves in the sample, pieces | | | | |
|---|---|---|---|---|---|
| | without MF | $H = 0,5$ кА/м | $H = 1$ кА/м | $H = 2$ кА/м | $H = 4$ кА/м |
| 0 | 20±0,4 | 20±0,4 | 20±0,4 | 20±0,4 | 20±0,4 |
| 24 | 20±0,4 | 20±0,4 | 20±0,4 | 20±0,4 | 20±0,4 |
| 72 | 26±0,4 | 21±0,4 | 23±0,4 | 26±0,4 | 22±0,4 |
| 144 | 33±0,5 | 21±0,4 | 25±0,4 | 35±0,5 | 23±0,4 |
| 216 | 40±0,6 | 22±0,5 | 28±0,5 | 44±0,6 | 25±0,4 |
| 288 | 45±0,6 | 22±0,5 | 30±0,5 | 53±0,6 | 26±0,5 |
| 360 | 60±0,7 | 23±0,5 | 35±0,5 | 62±0,7 | 30±0,5 |
| 432 | 75±0,8 | 24±0,5 | 36±0,6 | 70±0,7 | 35±0,5 |
| 480 | 80±0,8 | 24±0,5 | 45±0,6 | 75±0,8 | 44±0,6 |

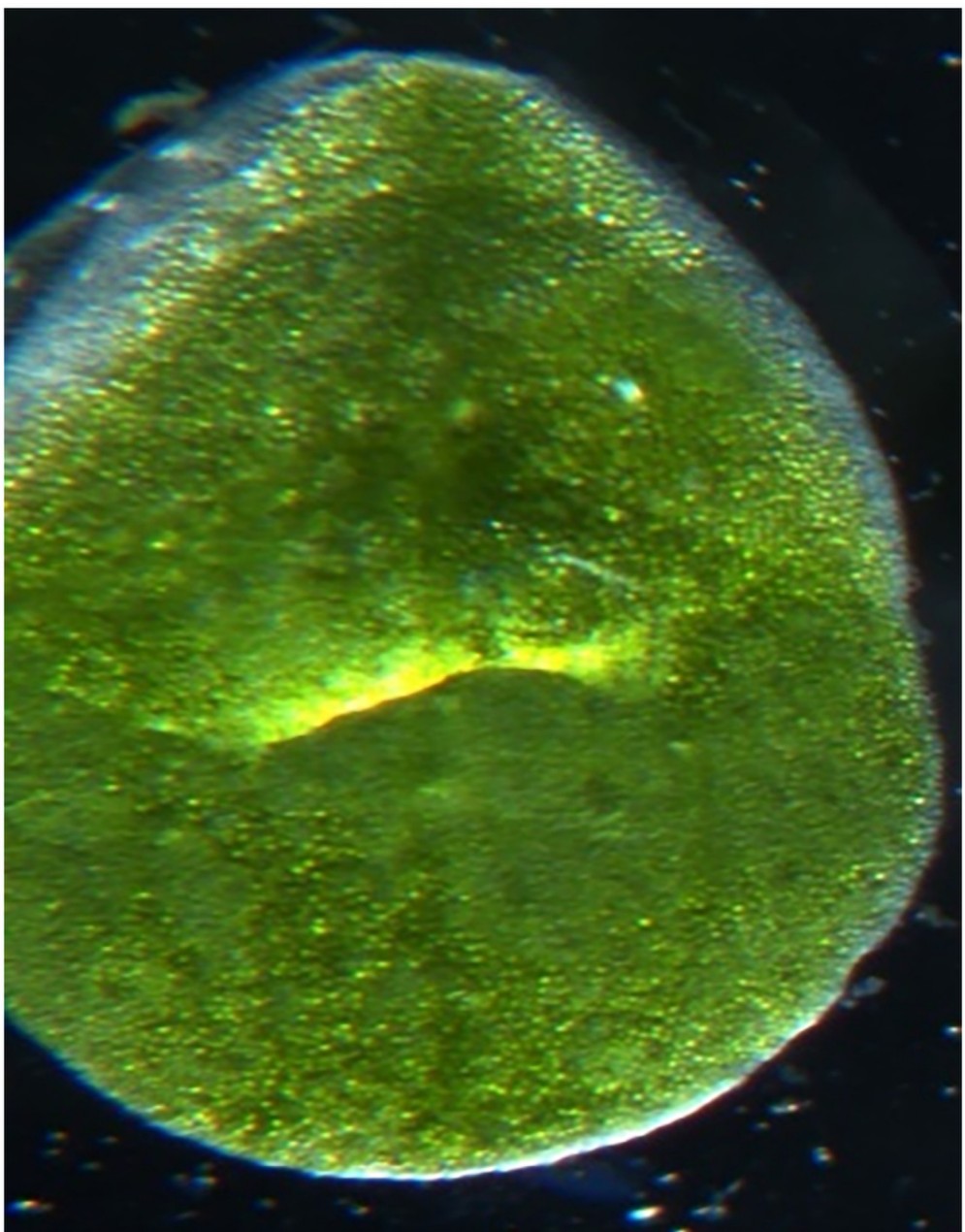

**Fig 4. Photo of *Lemna minor* duckweed in natural conditions.**

sixth day the leaves became brown-yellow, on the 15th day the duckweed completely turned brown and lost all signs of vitality, i.e. necrosis occurred. At $H = 4.0$ kA/m the *Lemna minor* duckweed actively degraded the structure of plant material, the water penetrated into the phytomass, the plant color changed from bright green, in the initial state, to a brown color during the period of inhibition. Being further influenced for 10 days, the plant died (Fig 5).

At the next stage the combined effect of magnetic fields and weak electric fields on the processes of the copper extraction from the sulfate solutions by duckweed *Lemna minor* was studied. Plants were placed in an electrochemical cell (with an aluminum cathode and a graphite

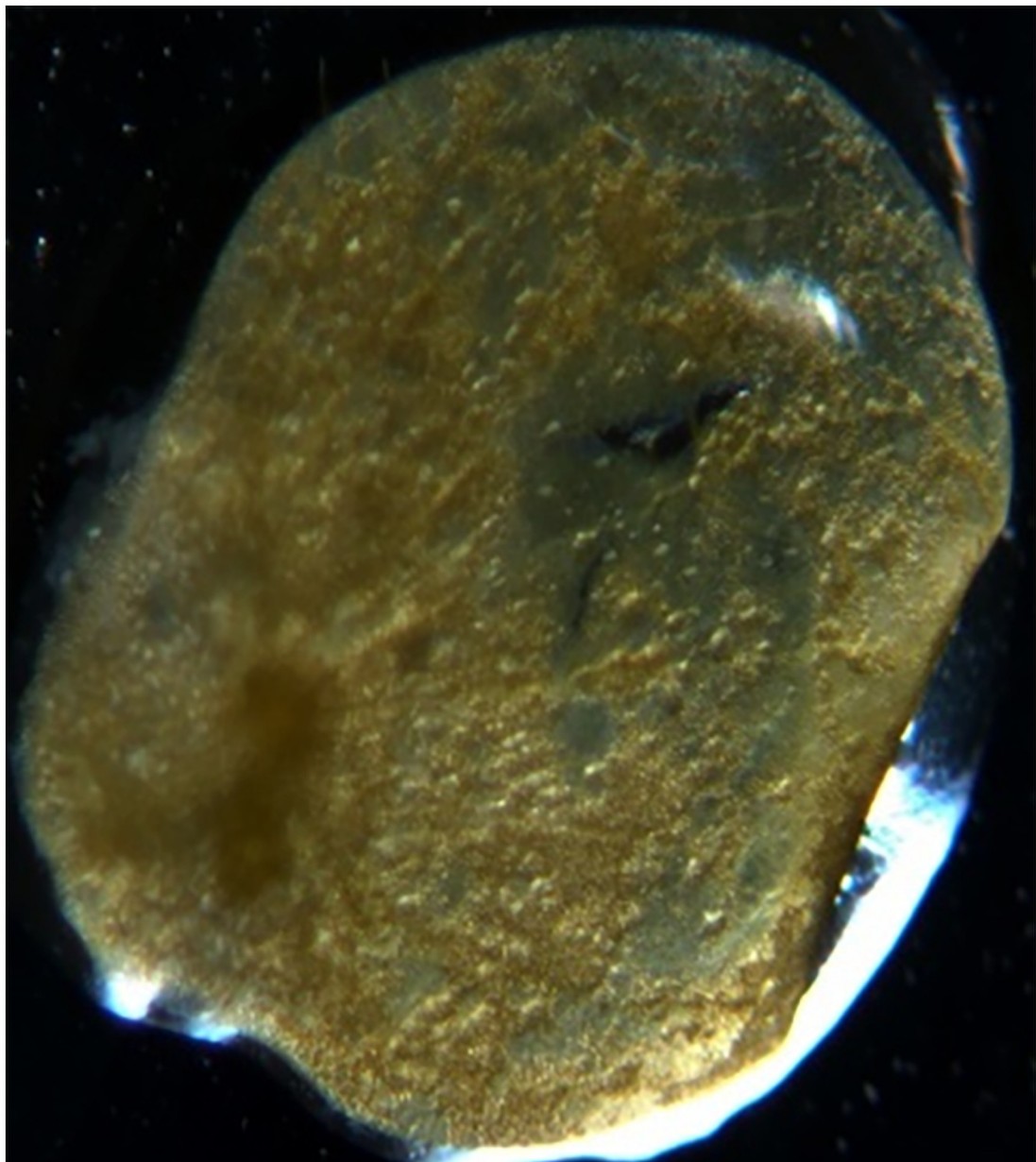

**Fig 5. Photo of *Lemna minor* duckweed aged for 10 days in a solution of $CuSO_4$ with concentration $C_i$ = 1 mg/L when influenced to MF $H$ = 4 kA/m.**

anode located in a working solution of $CuSO_4$) at given current densities $j$ = 80, 240, 480 µA/cm². The cell was placed in the facility creating a magnetic field with $H$ = 2 kA/m as a value having the best effect of phytoremediation processes.

The obtained data on the influence of current density on the process of extracting copper duckweed made it possible to establish that the rate of water purification from copper cations increases with the increasing of the time duration of the plant in solution (Table 2). The minimum concentration and maximum rate of phytosorption of copper are achieved in the first hours of being the duckweed in a solution. The dependence is repeated regardless of the magnitude of the current density, at the same time, the extraction rate depends on these indicators.

**Table 2. Residual concentration (C), purification efficiency Pe, extraction velocity (Ve) of copper cations from a *CuSO4* solution ($C_{initial}$ = 1 mg/L $Cu^{2+}$) by duckweed vs current densities (j) (H = 2 kA/m).**

| Current density j, μA/cm$^2$ | Time t, h | Residual concentration C, mg/L | Extraction velocity Ve, mg/h | Pe, % |
|---|---|---|---|---|
| 0 | 1 | 0,84+0,04 | 0,16 | 16 |
| | 3 | 0,78+0,04 | 0,07 | 22 |
| | 5 | 0,79+0,03 | 0,06 | 21 |
| | 24 | 0,80+0,03 | 0,02 | 20 |
| 80 | 1 | 0,71+0,04 | 0,28 | 29 |
| | 3 | 0,61+0,03 | 0,12 | 39 |
| | 5 | 0,65+0,03 | 0,07 | 35 |
| | 24 | 0,69+0,03 | 0,01 | 31 |
| 240 | 1 | 0,64+0,04 | 0,35 | 36 |
| | 3 | 0,53+0,03 | 0,15 | 47 |
| | 5 | 0,53+0,04 | 0,11 | 47 |
| | 24 | 0,55+0,01 | 0,03 | 45 |
| 480 | 1 | 0,70+0,03 | 0,29 | 30 |
| | 3 | 0,44+0,02 | 0,18 | 56 |
| | 5 | 0,50+0,02 | 0,09 | 50 |
| | 24 | 0,59+0,02 | 0,02 | 41 |

The lowest extraction velocity was achieved at a current density of $j$ = 80 μA/cm$^2$. The highest cleaning efficiency (up to 47%) was achieved by holding the duckweed in a *CuSO4* solution at a current density of $j$ = 240 μA/cm$^2$. At a higher current ($j$ = 480 μA/cm$^2$) the cleaning effect decreased, which is apparently due to the physical state of the duckweed which changed its external characteristics during the biosorption. The plant lost its bright green color and became lighter. At the same time, we observed a more efficient extraction of copper by the duckweed. After holding the plant in the solution for 5 h, the excess copper concentration discharge into the solution has been achieved. A subsequent copper recovery occurred at lower rates.

Our results on the effect of an electric current are consistent with the published data [62,63]. It is known that plant cell membranes have the ability to concentrate electric fields. When an additional external field is applied to the cells, the conductivity of cell membranes sharply increases [63]. After moderate electric treatment (in our case, it is $j$ = 80 μA/cm$^2$) the cell conductivity and, consequently, the biosorption process gradually decrease. With more intense electrical treatment ($j$ = 480 μA/cm$^2$) an irreversible destruction of part of the cells occurs, which worsens the biosorption capacity of the membranes and often leads to the electrical breakdown. The electric current $j$ = 240 μA/cm$^2$ is close to the biorhythm current, therefore, it contributes to the increase in cell permeability and has a beneficial effect on the biosorption of metal cations [62]. Under the influence of the electric current single electric pores appear in the cell membrane, which can change their sizes [63]. The processes associated with changes in the cell membrane under the influence of an electric field are called electroporation. During electroporation a local restructuring occurs in the membrane, leading to the appearance of a water channel through which micro and macro particles can travel at high speeds [64]. Large molecules are able to expand pores which then slowly (~ 100 s) relax to the initial state.

The microstructural analysis of duckweed leaves subjected to physical stresses carried out by using an electron scanning microscope. Microstructural of duckweed leaves *Lemna minor* without influence in natural condition is shown in Fig 6. The duckweed leaves with the

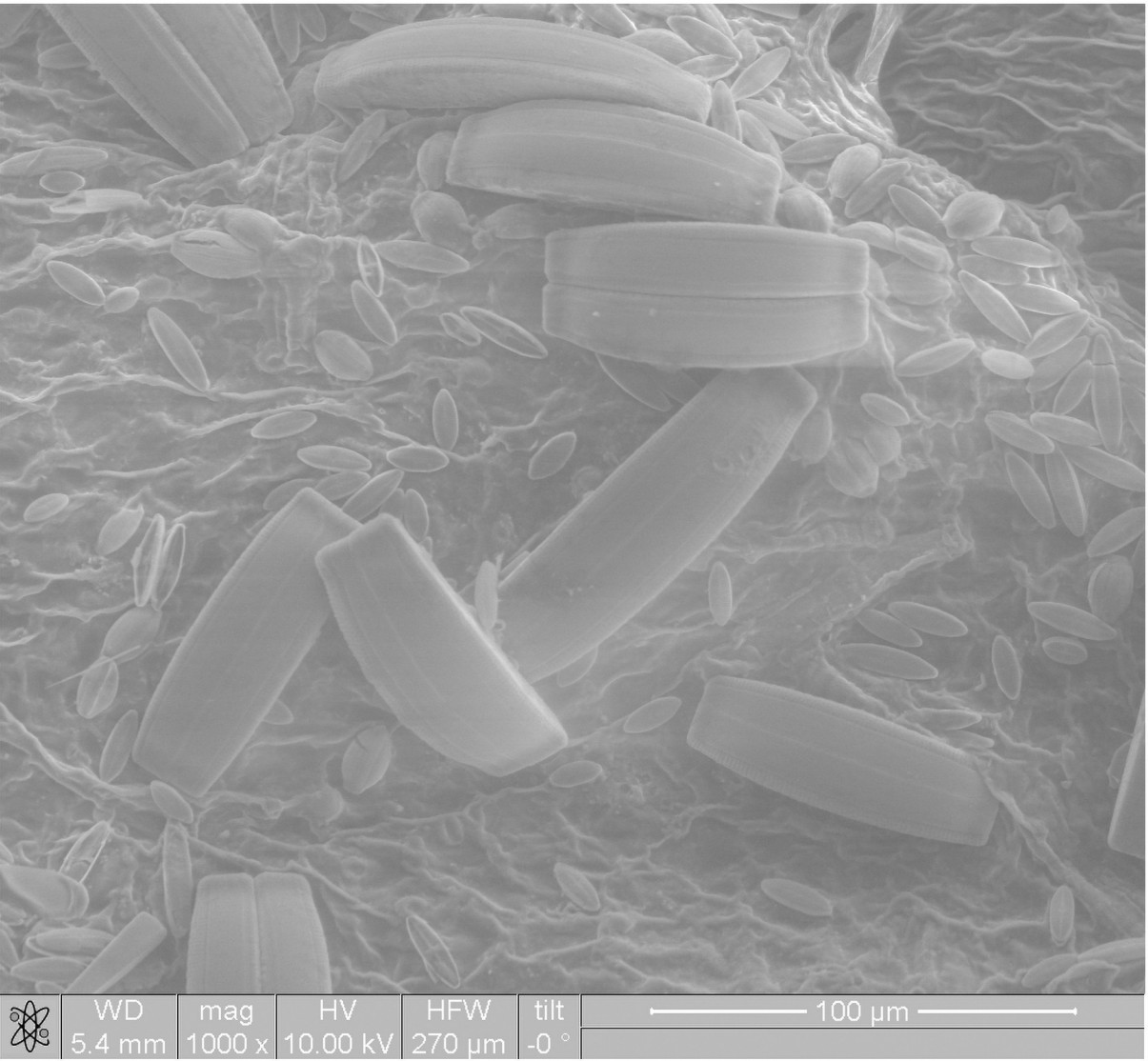

**Fig 6. Microstructural of duckweed leaves *Lemna minor* without influence in natural condition.**

combined influence of a magnetic field and an electric field in the $CuSO_4$ solution is shown in Fig 7. An analysis if these figures made it possible to establish that the structure of duckweed leaves surface changes in the $CuSO_4$ solution compared to the initial one. In the cell structure under the influence of MF ($H$ = 2 kA/m) and weak electric fields ($j$ = 240 μA/cm$^2$) the watering of the plant tissues has been occurred.

The main problem of biosorption treatment is the regeneration and utilization of biosorbents. In the case of bioobjects regeneration is not possible, therefore waste plants must be disposed of. Since there is no excessive accumulation of hazardous quantities of harmful substances in the plant phytomass, it can be used for paper and biofertilizers, processing for gas and liquid fuel [27]. Knowing the maximum permissible concentration (MPC) values of the content of heavy metals in animal feed (for example, in Russian Federation MPC for $Zn$ = 50 mg/kg; MPC for $Cd$ = 0.3 mg/kg; MPC for $Cu$ = 10 mg/kg; MPC for $Fe$ = 0.1 mg/kg), we can calculate the daily dose of spent phytosorbents as feed. But often waste plants contain

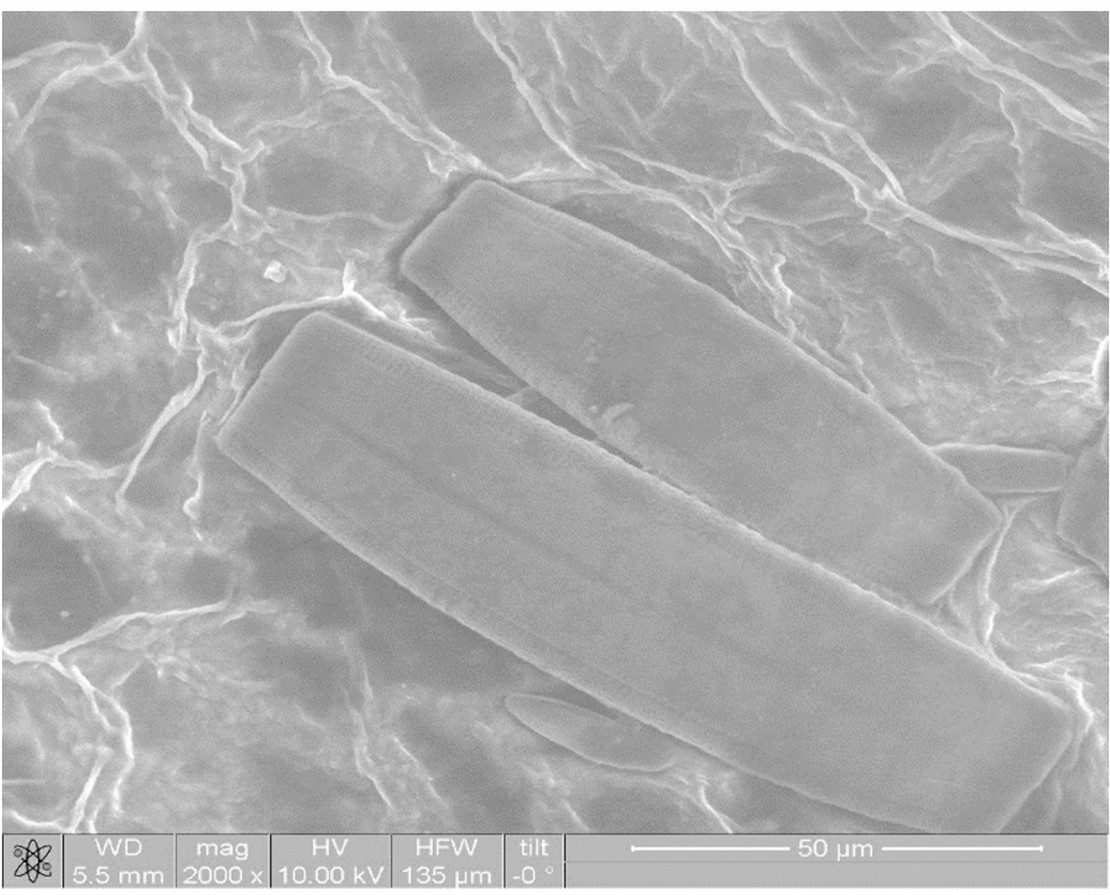

**Fig 7. Microstructural of duckweed leaves *Lemna minor* with the combined influence of a magnetic field and an electric field in the *CuSO₄* solution.**

toxic components that exceed the permissible value, in this case other methods of disposal are necessary.

Electrochemical extraction was carried out on the duckweed, which was previously placed into a solution of copper sulfate as described above. The combined influence of a MF (intensity $H = 2$ kA/m) and weak electric fields with different current densities ($j = 80, 240, 480$ μA/cm$^2$) was investigated. The experimental results (Fig 8) made it possible to establish that the highest copper content in the phytomass of the cassock plant was found during the phytosorption of copper by the plant under the combined influence of MF $H = 2$ kA/m and the direct current density of $j = 240$ μA/cm$^2$. This confirmed the presented data on the influence of weak electric fields on the enhancement of a metal sorption by the phytosorbents.

According to the formulas (1) and (2), the mass of pure copper released on the electrode was calculated. The mass was determined by the potentiostatic method (PS) for a comparison with the mass of copper absorbed by the plant from the solution during phytoremediation, determined by the voltammetric method (VA) (Table 3), was made. In Table 3 comparative results of VA and PS methods for the determination of copper extracted from *Lemna minor* duckweed influenced to MF (H = 2 kA/m) and weak electric fields (j, μA/cm$^2$) for 144 hours are presented. In Table 3 The results are presented in mg per 100 g of phytosorbent for the following options: copper release potentials for PS method for 1 –E = 0.30 V; for 2 –E = 0.32 V; for 3 –E = 0.34 V.

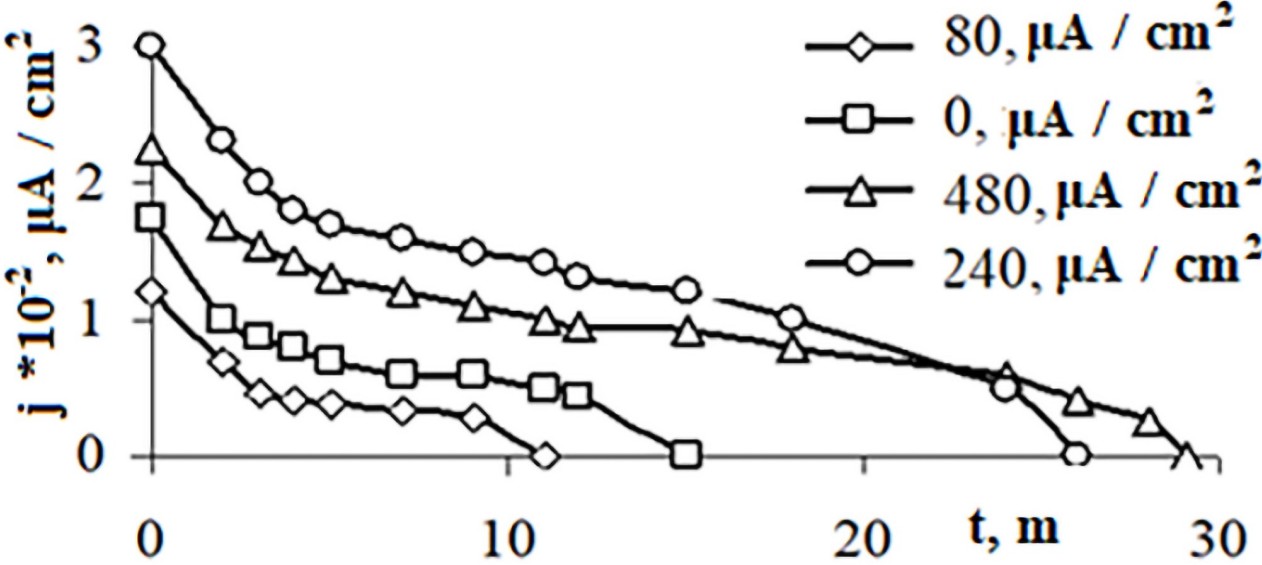

**Fig 8. Potentiostatic curves obtained when copper was isolated from the eluate of the spent phytomass of *Lemna minor* duckweed at *E* = 0.32 V.** The duckweed was previously placed into a copper sulfate solution with $Cu^{2+}$ ($Ci$ = 1 mg/L) under the influence of MF = 2 kA/m, at various current densities $j$, µA/cm$^2$.

Table 3 shows that the extraction of copper from the eluate of spent phytomass practically coincides with the mass of copper absorbed by the plant in the process of phytomass. The maximum metal release occurs at *E* = 0.32V. The developed electrochemical method of utilization of waste phytomass makes it possible to extract the metal almost completely. The recovered metal can be reused in the national economy, that will save natural resources. This corresponds to the principles of a circular economy of zero waste production, which is considered as a new business model for a more sustainable development of society [65].

## 4. Technological recommendations

To use phytoremediation processes on the basis of bioeconomics, technological recommendations have been developed and a scheme for WW treatment from copper by using *Lemna*

**Table 3. Comparative results of VA and PS methods.**

| Parameter, $j$, µA/cm$^2$ | Copper release potentials for PS method | Research Methods | |
|---|---|---|---|
| | | VA, mg per 100 g | PS, mg per 100 g |
| | 1 | 0,233± 0,002 | 0,232± 0,002 |
| 0.00 | 2 | 0,215± 0,002 | 0,213± 0,002 |
| | 3 | 0,239± 0,002 | 0,230± 0,002 |
| | 1 | 0,325± 0,002 | 0,273± 0,002 |
| 80 | 2 | 0,271± 0,003 | 0,266± 0,002 |
| | 3 | 0,294± 0,003 | 0,283± 0,003 |
| | 1 | 0,381± 0,003 | 0,320± 0,003 |
| 240 | 2 | 0,385± 0,004 | 0,312± 0,003 |
| | 3 | 0,372± 0,004 | 0,311± 0,003 |
| | 1 | 0,241± 0,002 | 0,238± 0,002 |
| 480 | 2 | 0,240± 0,002 | 0,238± 0,002 |
| | 3 | 0,234± 0,002 | 0,230± 0,002 |

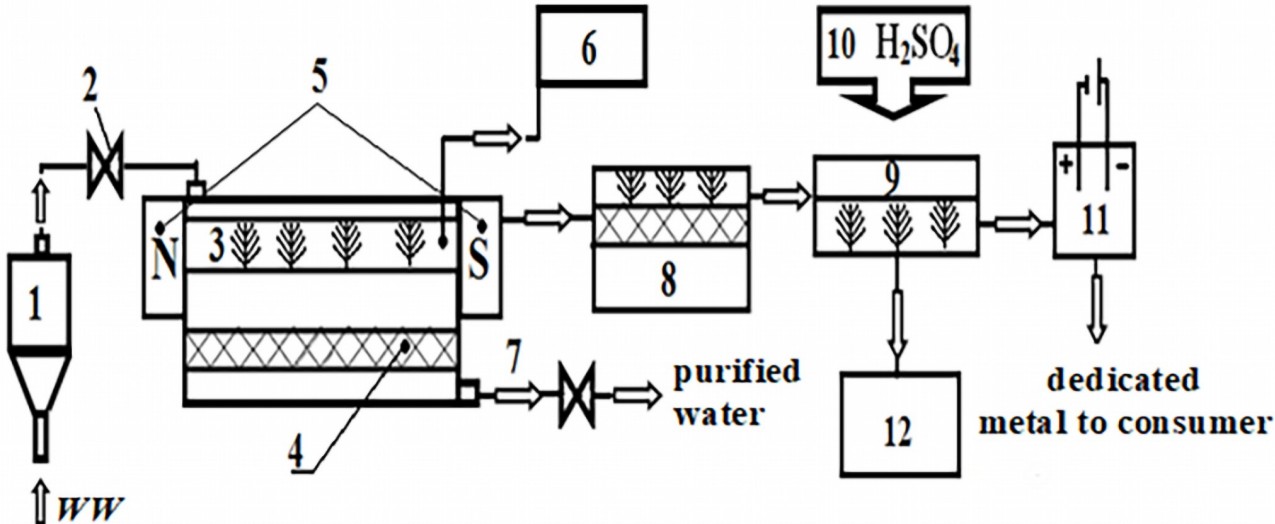

**Fig 9. Schematic diagram of the phytoremediation of metals from WW.** 1 –a WW averager; 2 –a pump for pumping averaged water into a biopond; 3 –a biopond populated by higher aquatic vegetation; 4 –a mesh pan for collecting and holding *Lemna minor* duckweed during the discharge of purified water; 5 –an installation of a constant magnetic field (can be combined with weak electric fields.); 6 –an equipment for monitoring the residual content of copper cations in solution; 7 –a pump for pumping purified water; 8 –lifting the waste mass by a pallet; 9 –bath for the preparation of the eluate; 10 –a dispenser of sulfuric acid; 11 –a metal electrolysis bath; 12 –a capacity for the collection of neutralized phytomass and its disposal.

*minor* duckweed and physical influences has been proposed. The scheme of the proposed technology is presented in the Fig 9.

The results of the research made it possible to formulate the following technological recommendations:

1. *Lemna minor* duckweed is taken from natural conditions or grown in previously defended water at least until the ripening period (leaf size—5. . . 6 mm). The plant should have a bright green color.

2. *Lemna minor* duckweed is transferred to the biopond with WW in an amount of 1.0–1.5 kg/m$^3$ of WW with an initial concentration, $Ci$ of $Cu^{2+}$ 1 cation of 5 mg/L.

3. The further cleaning process is carried out under selected optimal conditions for influence to fields of various nature:

   - Without physical effects the duckweed is kept in drains for 12–15 days to the required degree of treatment of WW.

   - When influenced to MF with a field strength of $H = 2.0$ kA/m, the plant is kept in WW for 8 days.

   - When influenced to magnetic fields with a field strength of $H = 2.0$ kA/m together with a weak electric field $j = 240$ μA/cm$^2$, the plant is kept in the WW for 6 days.

4. In the process of phytoremediation the residual content of copper ions in the solution is controlled. The process is carried out to the maximum permissible concentration of copper ions in the MPC $Cu^{2+}$ solution = 0.1 mg/L—a toxicological indicator for water bodies.

5. A regular visual inspection of plants is done. When the color changes to pale green or brown (signs of cytoplasmolysis and necrosis), the duckweed is removed from the biological pond and replaced with new plants.

6. Pure copper is isolated from the spent biomass by electrochemical method and ready for further use.

This technology has several advantages: high cleaning efficiency, low cost and low energy consumption, versatility, environmental safety, ease of implementation, non-waste.

## 5. Conclusions

It was found that the removal of copper cations by duckweed from sulfate solutions most effectively proceeds in the initial period of time (1–5 hours) and under the influence of a magnetic field with a strength of $H = 2$ kA/m. Studies of the growth and development purification of duckweed leaves under the influence of a magnetic field and without it have shown that the greatest growth and development of *Lemna minor* duckweed occurred without exposure and under the influence of a magnetic field with a strength of $H = 2$ kA/m. Tissue necrosis of *Lemna minor* duckweed fronds occurs after 10 days of being in sulfate copper solutions.

In the study of the combined effect of the magnetic field ($H = 2$ kA/m) and weak electric fields, it was shown that the highest extraction rate of copper cations is achieved when the duckweed is in the $CuSO_4$ solution at a current density of $j = 240$ μA/cm$^2$.

For the utilization and reuse of the spent phytomass the electrochemical release of copper from the eluate of the spent phytomass was proposed. The maximum metal release occurs at $E = 0.32$ V.

For the use of phytoremediation processes the technological recommendations and a wastewater treatment scheme have been developed in accordance with the principle of bioeconomics. This scheme allows achieving waste-free technologies and a closed cycle of using duckweed for environmental purposes. The introduction of this technology will contribute to the development of a "green economy".

## Author Contributions

**Conceptualization:** Natalia Politaeva.

**Data curation:** Natalia Politaeva.

**Methodology:** Natalia Politaeva.

**Supervision:** Vladimir Badenko.

**Validation:** Vladimir Badenko.

**Writing – original draft:** Natalia Politaeva.

**Writing – review & editing:** Vladimir Badenko.

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
