## [Decision Letter · Decision Letter 0]

10 May 2021

PONE-D-21-09527

Effect of Magnetic and Electric Fields on Acceleration of Phytoremediation Process Using Lemna Minor Duckweed for Extraction Copper Cations from Solutions

PLOS ONE

Dear Dr. Badenko,

Thank you for submitting your manuscript to PLOS ONE. After careful consideration, we feel that it has merit but does not fully meet PLOS ONE’s publication criteria as it currently stands. Therefore, we invite you to submit a revised version of the manuscript that addresses the points raised during the review process.

We look forward to receiving your revised manuscript.

Kind regards,

Muhammad Rizwan

Academic Editor

PLOS ONE

Journal Requirements:

2.Thank you for stating the following in the Acknowledgments Section of your manuscript:

"The research is partially funded by the Ministry of Science and Higher Education of the

Russian Federation as part of World-class Research Center program: Advanced Digital

Technologies (contract No. 075-15-2020-934 dated 17.11.2020)"

Reviewers' comments:

Reviewer's Responses to Questions

**Comments to the Author**

1. Is the manuscript technically sound, and do the data support the conclusions?

Reviewer #1: Partly

Reviewer #2: Yes

2. Has the statistical analysis been performed appropriately and rigorously? 

Reviewer #1: No

Reviewer #2: No

3. Have the authors made all data underlying the findings in their manuscript fully available?

Reviewer #1: Yes

Reviewer #2: Yes

4. Is the manuscript presented in an intelligible fashion and written in standard English?

Reviewer #1: Yes

Reviewer #2: No

5. Review Comments to the Author

Reviewer #1: The abstract section is not appropriate. The authors have not included the results of the study. Presently, the Abstract section is just an introduction to the study. I would advise organizing it the following way.

1: introductory sentences (1-2 sentences); 2: methodology (1-2 sentences); 3: description of the numerical results must be the main body of the Abstract (5-10 sentences); 4: conclusion (1-2 sentences. )

It's 'Keywords' and not the 'Key words'.

Arrange your keywords alphabetically

The authors didn't provide any references for the adopted experimental protocols/procedures. Moreover, the provided details are not enough to reproduce the study. Please, include the complete details of the methods with proper reference citations.

There is no statistical analysis for Table 1.

I don't know what has happened to Figure 1. There is just a Table and no figure(s).

Statistical analysis for the data presented in Table 2 is not sufficient. Please, compare it for significant differences.

No figure in Figure 5.

Reduce the conclusion section up to 5-8 lines.

Replace the references older than 10 years. You can include a reference older than 10 years if it is imperative to mention it due to exceptional circumstances.

Be selective and representative while citing the studies. A regular article should have 40-50 citations.

Please consider the following if you submit the revised version of the study to the journal or elsewhere.

1: must include line numbers

2: improve the quality of the pictures to highlight the main object/finding in it

3: arrange your figure to develop pictorial plates. (please, see the published papers to see how the pictures are arranged into pictorial plates)

Reviewer #2: The research presented in the manuscript is of significant quality. However, its presentation can be further improved by analyzing the data statistically. Moreover, English language editing is recommended. Few suggestions and comments are given as track changes.

6. PLOS authors have the option to publish the peer review history of their article (what does this mean?). If published, this will include your full peer review and any attached files.

Reviewer #1: No

Reviewer #2: **Yes: **Muhammad Baqir Hussain

---

## [Author Response · Author response to Decision Letter 0]

21 Jun 2021

Journal Requirements:

Response:

We have edited our manuscript, and we hope now our manuscript meets PLOS ONE's style requirements, including those for file naming.

2.Thank you for stating the following in the Acknowledgments Section of your manuscript:

"The research is partially funded by the Ministry of Science and Higher Education of the

Russian Federation as part of World-class Research Center program: Advanced Digital

Technologies (contract No. 075-15-2020-934 dated 17.11.2020)"

Response:

We have removed any funding-related text from the manuscript

We would like to update our Funding Statement as following: "The research is partially funded by the Ministry of Science and Higher Education of the Russian Federation as part of World-class Research Center program: Advanced Digital Technologies (contract No. 075-15-2020-934 dated 17.11.2020)"

We also have included amended statement within our cover letter.

Reviewers' comments:

Reviewer's Responses to Questions

Comments to the Author

1. Is the manuscript technically sound, and do the data support the conclusions?

Reviewer #1: Partly

Reviewer #2: Yes

Response:

We have partially improved the manuscript from this point of view

2. Has the statistical analysis been performed appropriately and rigorously?

Reviewer #1: No

Reviewer #2: No

Response:

We have improved the manuscript from this point of view. We have improved Table 1 and Table 3. It was found that critical value of Student's criterion = 95% for our experimental data.

3. Have the authors made all data underlying the findings in their manuscript fully available?

Reviewer #1: Yes

Reviewer #2: Yes

Response:

According reviewer answers, there is nothing to improve.

4. Is the manuscript presented in an intelligible fashion and written in standard English?

Reviewer #1: Yes

Reviewer #2: No

Response:

We have partially improved the manuscript from this point of view. Small corrections are highlighted by red color of the text and otherwise corrections are highlighted using yellow background.

5. Review Comments to the Author

Reviewer #1: The abstract section is not appropriate. The authors have not included the results of the study. Presently, the Abstract section is just an introduction to the study. I would advise organizing it the following way.

1: introductory sentences (1-2 sentences); 2: methodology (1-2 sentences); 3: description of the numerical results must be the main body of the Abstract (5-10 sentences); 4: conclusion (1-2 sentences. )

Response:

We have completely rewritten the abstract according recommendation

It's 'Keywords' and not the 'Key words'.

Arrange your keywords alphabetically

Response:

We have rewritten the keywords

The authors didn't provide any references for the adopted experimental protocols/procedures. Moreover, the provided details are not enough to reproduce the study. Please, include the complete details of the methods with proper reference citations.

There is no statistical analysis for Table 1.

Response:

We have added some additional explanation in the text

I don't know what has happened to Figure 1. There is just a Table and no figure(s).

Response:

All figures for the manuscript have been improved. All figures have been checked in Preflight Analysis and Conversion Engine (PACE) digital diagnostic tool, https://pacev2.apexcovantage.com/.

Statistical analysis for the data presented in Table 2 is not sufficient. Please, compare it for significant differences.

Response:

Table 2 have been edited

No figure in Figure 5.

Response:

All figures for the manuscript have been improved. All figures have been checked in Preflight Analysis and Conversion Engine (PACE) digital diagnostic tool, https://pacev2.apexcovantage.com/.

Reduce the conclusion section up to 5-8 lines.

Response:

We have completely rewritten the conclusions section

Replace the references older than 10 years. You can include a reference older than 10 years if it is imperative to mention it due to exceptional circumstances.

Response:

We have replaced old references (see yellow background in References section)

Be selective and representative while citing the studies. A regular article should have 40-50 citations.

Response:

We have reduced the number of references

Please consider the following if you submit the revised version of the study to the journal or elsewhere.

1: must include line numbers

Yes, we have included line numbers

2: improve the quality of the pictures to highlight the main object/finding in it

All figures for the manuscript have been improved. All figures have been checked in Preflight Analysis and Conversion Engine (PACE) digital diagnostic tool, https://pacev2.apexcovantage.com/.

3: arrange your figure to develop pictorial plates. (please, see the published papers to see how the pictures are arranged into pictorial plates)

All figures for the manuscript have been improved. All figures have been checked in Preflight Analysis and Conversion Engine (PACE) digital diagnostic tool, https://pacev2.apexcovantage.com/.

Reviewer #2: The research presented in the manuscript is of significant quality. However, its presentation can be further improved by analyzing the data statistically. Moreover, English language editing is recommended. Few suggestions and comments are given as track changes.

Response:

We have partially improved the manuscript from this point of view. Small corrections are highlighted by red color of the text and otherwise corrections are highlighted using yellow background.

6. PLOS authors have the option to publish the peer review history of their article (what does this mean?). If published, this will include your full peer review and any attached files.

Do you want your identity to be public for this peer review? For information about this choice, including consent withdrawal, please see our Privacy Policy.

Reviewer #1: No

Reviewer #2: Yes: Muhammad Baqir Hussain

These comments do not require a response.

---

## [Decision Letter · Decision Letter 1]

19 Jul 2021

Magnetic and Electric Field Accelerate Phytoextraction of Copper Lemna Minor Duckweed

PONE-D-21-09527R1

Dear Dr. Badenko,

We’re pleased to inform you that your manuscript has been judged scientifically suitable for publication and will be formally accepted for publication once it meets all outstanding technical requirements.

Kind regards,

Muhammad Rizwan

Academic Editor

PLOS ONE

Additional Editor Comments (optional):

Reviewers' comments:

Reviewer's Responses to Questions

**Comments to the Author**

1. If the authors have adequately addressed your comments raised in a previous round of review and you feel that this manuscript is now acceptable for publication, you may indicate that here to bypass the “Comments to the Author” section, enter your conflict of interest statement in the “Confidential to Editor” section, and submit your "Accept" recommendation.

Reviewer #1: All comments have been addressed

Reviewer #2: All comments have been addressed

2. Is the manuscript technically sound, and do the data support the conclusions?

Reviewer #1: Yes

Reviewer #2: Yes

3. Has the statistical analysis been performed appropriately and rigorously? 

Reviewer #1: Yes

Reviewer #2: Yes

4. Have the authors made all data underlying the findings in their manuscript fully available?

Reviewer #1: Yes

Reviewer #2: Yes

5. Is the manuscript presented in an intelligible fashion and written in standard English?

Reviewer #1: Yes

Reviewer #2: Yes

6. Review Comments to the Author

Reviewer #1: I verify that all required questions have been answered and that all responses meet formatting specifications. I recommend ACCEPTANCE of the current version.

Reviewer #2: The suggestions made in previous review have been incorporated very well. Now the manuscript may be accepted for publication in present form. However, a careful re-read of the manuscript is recommended by the authers to avoid any mistake text, data and references.

7. PLOS authors have the option to publish the peer review history of their article (what does this mean?). If published, this will include your full peer review and any attached files.

Reviewer #1: No

Reviewer #2: **Yes: **Muhammad Baqir Hussain

---

## [Editor Report · Acceptance letter]

22 Jul 2021

PONE-D-21-09527R1 

Magnetic and Electric Field Accelerate Phytoextraction of Copper *Lemna Minor* Duckweed 

Dear Dr. Badenko:

I'm pleased to inform you that your manuscript has been deemed suitable for publication in PLOS ONE. Congratulations! Your manuscript is now with our production department. 

Kind regards, 

on behalf of

Dr. Muhammad Rizwan 

Academic Editor

PLOS ONE